# PARP Inhibitor Inhibits the Vasculogenic Mimicry through a NF-κB-PTX3 Axis Signaling in Breast Cancer Cells

**DOI:** 10.3390/ijms232416171

**Published:** 2022-12-18

**Authors:** Justine Chivot, Nathalie Ferrand, Aude Fert, Patrick Van Dreden, Romain Morichon, Michèle Sabbah

**Affiliations:** 1Team Cancer Biology and Therapeutics, Centre de Recherche Saint-Antoine (CRSA), Inserm UMR_S 938, Institut Universitaire de Cancérologie, Sorbonne University, 75012 Paris, France; 2Clinical Research Department, Diagnostica Stago, 92230 Genevilliers, France; 3CRSA Cytométrie Imagerie Saint-Antoine, Sorbonne University, 75012 Paris, France; 4Centre National de la Recherche Scientifique (CNRS), 75016 Paris, France

**Keywords:** PARP inhibitor, vasculogenic mimicry, PTX3, NF-κB signaling

## Abstract

Poly (ADP-ribose) polymerase inhibitors (PARPi) are targeted therapies that inhibit PARP proteins which are involved in a variety of cell functions. PARPi may act as modulators of angiogenesis; however, the relationship between PARPi and the vasculogenic mimicry (VM) in breast cancer remains unclear. To determine whether PARPi regulate the vascular channel formation, we assessed whether the treatment with olaparib, talazoparib and veliparib inhibits the vascular channel formation by breast cancer cell lines. Here, we found that PARPi act as potent inhibitors of the VM formation in triple negative breast cancer cells, independently of the BRCA status. Mechanistically, we find that PARPi trigger and inhibit the NF-κB signaling, leading to the inhibition of the VM. We further show that PARPi decrease the expression of the angiogenic factor PTX3. Moreover, PTX3 rescued the PARPi-inhibited VM inhibition. In conclusion, our results indicate that PARPi, by targeting the VM, may provide a new therapeutic approach for triple negative breast cancer.

## 1. Introduction

Breast cancer is the most commonly diagnosed cancer among women worldwide and is the leading cause of death by cancer, in women [1]. Breast cancer patients’ responses to hormonal and targeted therapies depend on the estrogen and progesterone receptors (ERs, PRs), along with the human epidermal growth factor receptor 2 (HER2) expression. Approximately 15% of breast tumors are totally devoid of these three receptors, harboring thus a triple-negative breast cancer (TNBC) phenotype [2]. In view of their specific behavior, compared to other subtypes, the treatment of TNBC remains challenging, due to the aggressiveness, increased risk of relapse, and the limited target therapies [3]. Several data support a central role for angiogenesis in the breast cancer growth and metastasis. However, the effect of anti-angiogenic monotherapy appears to be limited in breast cancer [4]. Furthermore, an angiogenesis-independent pathway, known as the vasculogenic mimicry (VM) has been reported in very aggressive tumors. VM was first reported in melanoma but is now described in many invasive and metastatic cancers [5]. This phenomenon describes the formation of a vascular-like network lined by tumor cells, rather than by endothelial cells, that express certain typical endothelial proteins [6]. Many TNBCs form vascular networks, and this process may play an important role in the resistance to anti-angiogenic treatments [7,8].

PARPi are small inhibitors of the poly-(ADP)-ribose polymerase (PARP) family of DNA repair enzymes [9,10]. Several PARPi have been approved by the Food Drug Administration (FDA), to treat different cancers. These drugs have shown a great efficacy in the treatment of BRCA1/2-mutated tumors, such as ovarian and breast cancers [11,12]. These molecules have an anti-tumor activity by inhibiting the PARP activity and act by operating on the principle of the synthetic lethality, and leads to the cytotoxicity in BRCA deficient cells [13]. However, there is strong evidence from the preclinical and mechanistic studies on PARPi, that support the use of PARPi, irrespective of the BRCA1/2 status or the HR-mediated repair in triple negative breast cancer, and seems highly promising for the HER2 + tumors, that become resistant to trastuzumab [14,15,16]. PARP proteins mediate many biological actions, in addition to DNA repair, they act as transcriptional regulators of inflammation, apoptosis, chromatin modification, hypoxic response, epithelial mesenchymal transition, autophagy, and cancer stem cell programming [17]. It was also observed that, apart from the PARP inhibition, some of these molecules target a variety of kinases implicated in the signal transduction pathways, in both healthy and malignant cells [18]. All of these processes are involved in the tumor progression and highlight the relevance of the use of PARPi as a therapeutic approach. The role of PARP-1 in promoting angiogenesis that fuels the growth of tumors, can also be a target of PARPi, because the PARP-1 depletion or PARPi reduce the vessel formation [19] and the expression of the markers of angiogenesis in melanoma [20] or endothelial cells [21]. Moreover, the anti-angiogenic properties of the PARP inhibitors have been demonstrated, both in vitro and in vivo. PARP1 is activated by a direct interaction with the phosphorylated ERK2 and the PARP inhibition causes the loss of the ERK2 stimulation, by decreasing the activity of the critical pro-angiogenic factors [19,20,21,22,23,24]. This ultimately results in the reduced angiogenesis and inflammation. Interestingly, a recent study showed that the progression free survival was superior in ovarian cancer patients, who received a combination of the PARPi Olaparib, combined with the humanized monoclonal antibody bevacizumab that inhibited VEGF, compared to bevacizumab alone [25]. Other factors are also involved in the mechanisms of tumor angiogenesis. The pentraxin 3 (PTX3) protein, a member of the pentraxin family, has many roles in the innate immune response, including inflammation, but also in the angiogenic processes. Several studies have shown that PTX3 can inhibit or promote angiogenesis, depending on the context and the tumor [26]. In particular, it can interact with members of the FGF family, including FGF2, and thus inhibit its pro-angiogenic action [27]. Conversely, the increase in PTX3 is correlated to glioblastoma, with an increase in IL-8 and VEGF, two important factors in angiogenesis [28].

The transcription factor NF-κB formed by p65-p50 proteins, regulates a number of cellular pathways and multiple aspects of oncogenesis. NF-κB in the cytoplasm is inhibited by IκB proteins and the phosphorylation of IκB leads to the NF-κB translocation in the nucleus and transcription of its target genes. An inhibitory role of the NF-κB signaling has been found in tumor angiogenesis [29]. Furthermore, responsiveness to olaparib in head and neck cancer cell lines is correlated to high NF-κB levels [30].

In the present study, we found that three different PARPi can inhibit the VM formation in triple negative breast cancer cell lines. Mechanistically, we uncovered that PARPi, by targeting the NF-κB signaling pathway, identify PTX3 as a new potential candidate involved in the VM formation.

## 2. Results

### 2.1. VM Formation in Triple Negative Breast Cancer Cells Is Independent of the BRCA Status

Numerous studies have highlighted the presence of a network of the VM in the most aggressive breast cancers. First, we screened a large panel of 15 established breast cancer cell lines, in order to identify those that form networks of matrix-rich tubular structures in a three-dimensional culture (Table 1). Although all luminal cell lines do not form the VM, the microscopy experiments revealed that the cell lines capable of forming the VM form channels in a similar manner to the endothelial HMEC-1 cells (Figure 1). Moreover, they are associated with the basal-like breast cancer subtype and thus independently of the BRCA1/2 status (Figure 1).

### 2.2. PARP Inhibition Suppresses the Vasculogenic Mimicry in Breast Cancer Cells

Because PARP-1 is involved in angiogenesis, we next investigate the impact of the PARP inhibitor (PARPi) on the VM formation of three breast tumor cell lines and an endothelial cell line. We focus here on three PARPis: olaparib (AZD2281), talazoparib (BMN 673), and veliparib (ABT-888). Olaparib and talazoparib have been approved by the FDA for treating advanced ovarian, pancreatic, and breast cancers, veliparib does not yet have an approved label; nevertheless, there are currently promising results available in preclinical and early clinical settings. These three PARPis have been shown to target the PARP1/2 activity [31,32,33]. The cells were treated for 72 h with non-cytotoxic doses of PARPis determined by staining with annexin (Appendix A). PARPi treatments inhibited the formation of the tubular network in both breast tumor cell lines and the endothelial cell line HMEC-1 (Figure 2 and Appendix A).

### 2.3. PARPi Treatment Caused the Decreased Expression of the Angiogenic Proteins

In order to evaluate the PARPi effect on the level of angiogenic molecules, the cell culture supernatants from SUM159 and HMEC-1 cells, treated with the vehicle or olaparib (20 µM) for 72 h, were tested on a human angiogenesis array containing 55 different angiogenic proteins. Several angiogenic proteins were down regulated by olaparib on the membrane arrays (Figure 3A). Interestingly, as shown in Figure 3, there were differences in both the constitutive and olaparib treated angiogenic protein expressions between the HMEC-1 and SUM159 cell lines. We found major differences in the IL-8, PTX3 and VEGF expressions when the SUM159 cells were treated with Olaparib, compared to HMEC-1. These results were confirmed by ELISA assays showing a significantly lower concentration of PTX3 and VEGF in SUM159, compared to HMEC-1, when treated separately with the three PARPis. However, the ELISA assays did not confirm the IL-8 decrease in the SUM159 cells treated with olaparib. As expected, VEGF was weakly expressed in the unstimulated HMEC-1.

### 2.4. NF-κB Inactivation Is Required for the PARPi-Inhibited VM Formation

PARP1 is known to interact with the nuclear factor-kappa B (NF-κB) [34], which induces the transcription of various genes, including those related to the production of angiogenic molecules and cytokines [29,35]. Therefore, we analyzed the mRNA levels of the NFκB-target genes regulated by the olaparib treatment in the SUM159 cells. A decreased expression was observed for the PTX3, VEGF, uPA, PAI1, TIMP1, and IL-8 genes when the cells are treated for 72 h with olaparib. Interestingly, thrombospondin-1 (THSB1), an important inhibitor of angiogenesis [36], was up-regulated in the SUM159 cells treated with olaparib (Figure 4A).

We next aimed to analyze the influence of olaparib on the level of the total p65 protein and its phosphorylated form. The results of a western blot analysis in the SUM159 cellular extracts indicated that the basal p65 levels correlated with the protein phosphorylation at serine 536. Conversely, when the SUM159 cells were treated with 20 µM olaparib, we detected a reduction of the total p65 and phospho-p65 levels (Figure 4A). We then investigated whether the induced expression of IκBα might also contribute to the olaparib-decrease of the p65 expression levels. We showed a significant increase of IκBα and a slight decrease of the activator IKKα (Figure 4B). To check the NF-κB activity, the SUM159 cells were transfected with a NF-κB-dependent reporter in a transient transfection assay. The cells were treated with TNF-α, a known activator of NF-κB, and exposed to BAY 11-7082, an inhibitor of NF-κB or olaparib. The luciferase activity was determined [37]. The NF-κB activity was significantly increased upon treatment with TNF-α, and decreased when the cells were pretreated with BAY 11-7082 or olaparib (Figure 4C). No effect on the basal NF-κB activity was observed when the cells were incubated with BAY 11-7082 or olaparib alone.

We then investigated the influence of the NF-κB signaling during the VM, by using BAY 11-7082. When the SUM159 cells were treated with the inhibitor, a reduced tube formation was observed in a similar manner to that observed with olaparib (Figure 4D).

### 2.5. A NF-κB-PTX3 Axis Is Required for the VM Inhibition by PARPi

We next wished to validate the NF-κB target genes that could be candidates involved in the PARPi VM inhibition. Interestingly, the reduced NF-kB activity with BAY 11-7082 in the SUM159 cells reduced the level of the PTX3 expression (Figure 5A). To study the role of PTX3 in this process, we treated the SUM159 cells at the same time with a recombinant human PTX3 and olaparib before they were plated on Matrigel. We found that the PTX3 treatment was able to significantly reduce the effect of olaparib on the VM formation, as compared to the olaparib treatment (Figure 5B and Appendix A).

## 3. Discussion

PARP inhibitors (PARPi) represent an important group of therapeutic molecules that are particularly used in BRCA mutated tumors. However, recent data showed that certain PARPi, in pre-clinical studies, could inhibit the proliferation and promote the death of triple negative breast cancer cells without BRCA mutations [16,38,39]. Angiogenesis is crucial for the tumor growth and some studies have shown that PARPi present an anti-angiogenic activity and may affect the endothelial cell function. However, the mechanism underlying the effects of PARPi on the vascular plasticity remains relatively unknown. In the present study, we provide evidence that PARPi could impair the VM formation to inhibit the tumor progression via its action to inhibit the NF-kB activity, which subsequently leads to a decrease of PTX3.

We have demonstrated that three different PARPis are known to target the PARP1/2 activity, olaparib, talazoparib, and veliparib, impair the VM formation of triple negative breast cancer cells, independently of the BRCA1/2 status. Furthermore, our data showed that the status of BRCA is not a predictive biomarker of the VM formation in the TNBC since some BRCA-mutant HCC1937 or BT549 cell lines could form a network and others, such as MDA-MB-436 and SUM1315MO2, do not.

Here, we observe that PARPi reduce the expression of the angiogenic factors in the endothelial and tumor cells and most of these factors are regulated by NF-kB. Our results suggest that PARPi could inhibit the NF-kB activity in breast cancer cells. Actually, the effect of PARPi on the NF-kB expression and activity, is a subject of controversy and seems to be dependent on the cell type. While the PARP-1 invalidation or olaparib treatment inhibits the expression and the phosphorylation of NF-kB in small-cell lung cancer, olaparib treatment activates NF-kB in acute myeloid leukemia blasts [40,41]. Moreover, the susceptibility of HER2+ breast cancer to PARPi has been observed and the molecular mechanisms are attributed to the inhibition of the NF-kB signaling [42]. Additionally, NF-κB is frequently activated in the TNBC and inhibition of the NF-κB activity suppresses the growth of TNBC cells [43]. The transcription factor NF-κB has been connected with multiple aspects of angiogenesis, but its role has not been investigated in the vasculogenic mimicry. In the present study, we provide evidence that the pharmacological inhibition of the NF-κB factor with BAY 11-7082 could impair the VM formation.

Pentraxin-3 (PTX3) is a member of the pentraxin family, which includes CRP, it is involved in the innate immune response, and it complements the activation and inflammation [44]. PTX3 in cancer can act as an oncosuppressor or a protumorigenic factor [26]. PTX3 has been suggested to play a significant role in cancer-related inflammation and was shown to be increased in lung, prostate, ovarian, liver, and pancreatic carcinomas [45,46,47,48,49]. In breast cancer, the PTX3 expression is correlated with the poorly differentiated aggressive triple negative breast cancer [50,51]. Interestingly, PTX3 was found to induce a NF-κB signaling pathway in invasive melanoma [52]. Moreover, the NF-κB activation is involved in the PTX3 expression [53,54]. Our results reveal that PTX3 could suppress the VM-inhibition induced by Olaparib, suggesting that PTX3 could act to induce the NF-κB signaling and thus inhibits the effect of olaparib.

Angiogenesis is crucial for the tumor growth and studies have demonstrated that PTX3 could be an inducer or an inhibitor of angiogenesis. The enhanced expression of PTX3 correlates with the IL8-VEGF signaling axis and increases the invasion in glioblastoma cells [28] whereas serum PTX3, IL-8, and VEGF levels decreased in gastric carcinoma [55]. In addition, high PTX3 levels have been observed in tumor endothelial cells [56]. This is consistent with our results showing a decrease of PTX3, IL-8, and VEGF in the SUM159 cells treated with olaparib.

Several studies showed a PARP-1 dependent activation of NF-κB, a major factor during inflammation. Proteins involved in the DNA repair can serve to promote the NF-κB transcription of pro-inflammatory genes. Similar to the inhibition of PARP1, the inhibition of BER glycosylase OGG1 inhibited the NF-κB activity [57,58], suggesting that the targeting proteins involved in the pro-inflammatory signaling could be effective to inhibit the VM. Although the NF-κB signaling mediates the immune response, the inflammation pathway can lead to a malignant phenotype, facilitating the escape from the immune surveillance [59]. Furthermore, the PARP inhibitors can also enhance the anticancer activity of PD-1/PD-L1 or the CTLA-4 inhibition, demonstrating that the immunogenic effects of PARP can also be exploited for cancer therapy [60,61].

Moreover, we have recently shown that inhibiting the Hippo/YAP/TAZ signaling could inhibit the VM formation [62]. A crosstalk between the Hippo-YAP and NF-κB signaling has been demonstrated [63]. In many cases, the Hippo-YAP signaling was activated by the NF-κB signaling and inflammatory responses. As the YAP/TAZ inhibitors are currently under development, a combination of these inhibitors with PARP1 could be a novel approach to inhibit the VM formation.

The relationship between the effects of PARP and the NF-kB signaling in the PTX3 regulation is pursued. As PARP1 and p65NF-kB have been shown to interact and modulate the gene expression [64], therefore investigating this interaction in the PTX3 gene expression and furthermore the inhibition of these two signaling pathways is of great importance to elucidate the mechanisms regulating the PTX3 functions. Moreover, some data combining PARPi and NF-kB, showed an increased efficacy with the combination of two drugs [65], and also could be observed in the VM process.

In conclusion, we identified a novel molecular mechanism requiring the NF-κB inhibition that may contribute to prevent the formation of VM by the PARP inhibitor. Our study provides a rationale for targeting the PTX3-mediated NF-κB by PARPi, as part of a novel therapeutic approach for triple negative breast cancer.

## 4. Materials and Methods

### 4.1. Cell Lines and Drug Treatment

Human breast cancer cells were purchased from the ATCC (American Type Culture Collections, VA, USA). MCF-7, ZR-75.1, HCC38, BT-20, MDA-MB-231, and HS578T were cultured in DMEM (Dulbecco modified Eagle’s medium) supplemented with 10% FBS (fetal bovine serum) and 1% penicillin-streptomycin. T47D, BT474, SKBR-3, MDA-MB-436, BT549, and HCC1937 were cultured in RPMI 1640 supplemented with 10% FBS and 1% penicillin-streptomycin. SUM1315MO2 were cultured in Ham’s F12 medium, supplemented with 5% heat-inactivated FBS, 1% penicillin-streptomycin, 10 mM HEPES, 5 µg/mL insulin, and 10 ng/mL human recombinant EGF. SUM149 and SUM159 were maintained in Ham’s F12 medium, with 5% heat-inactivated FBS, 1% penicillin-streptomycin, 10 mM HEPES, 5 µg/mL insulin, and 2 µg/mL hydrocortisone. Human dermal microvascular endothelial cells HMEC-1 were cultured in MCDB 131 medium supplemented with 10% FBS, L-Glutamine 1%, 10 ng/mL human recombinant EGF, and 1 µg/mL hydrocortisone.

Olaparib (AZD2281), veliparib (ABT-888), talazoparib (BMN 673), and BAY11-7082 were purchased from Euromedex (Souffelweyersheim, France) and reconstituted, as described by the producer. The cells were treated with olaparib at 20 µM, veliparib at 10 nM, and talazoparib at 20 µM. BAY11-7082 was used at a concentration of 2.5 µM.

### 4.2. Tube Formation Assay

The tube formation assays were performed, according to manufacturer’s instructions. µ-Slide Angiogenesis (Ibidi 81506, Biovalley, Illkirch-Graffenstaden, France) were used and pre-coated with 10 µL of Matrigel^®^ growth factor reduced, phenol red-free (#356231, Corning, New York, NY, USA) and were allowed to polymerize at 37 °C for at least 30 min. The cells (2 to 3 × 10^5^ cells/mL) were seeded into each well, in triplicate, and maintained in the appropriate medium supplemented with 2% (*v*/*v*) FBS. A time-lapse video was recorded for 24 h to follow the formation of the tubules, using an IX83 inverted microscope (Olympus, Tokyo, Japan) with a motorized stage and an equipped 37 °C 5% CO_2_ thermostat chamber (Pecon; CelVivo, Inc, Maryland USA). The images were analyzed using ImageJ software (1.53q) and the plugin Angiogenesis Analyzer (number 1.0c) [66].

### 4.3. Real-Time RT-PCR

Total RNA was extracted from all cell lines using the RNeasy^®^ Plus Micro Kit (Qiagen, Hilden, Germany). The RNA quantity and purity were determined using a Spectrophotometer DS-11 (Denovix, Wilmington, DE, USA). One microgram of total RNA from each sample was reverse transcribed, and the real-time RT-PCR measurements were performed, as described previously [67], using an apparatus Aria MX (Agilent Technologies, Santa Clara, CA, USA) with the corresponding SYBR^®^ Green kit, according to the PROMEGA manufacturer’s recommendations. The data were analyzed using the comparative threshold cycle 2^−ΔΔCt^ method; the expression of 36B4 (RPL0) and β-actin were used to normalize the data (Table 2). The expression of the genes was expressed as the fold change in the samples, compared to the control condition. All values are averages of at least three independent experiments carried out in duplicate. The gene specific primers were purchased from Sigma Aldrich-Merck (Saint Louis, MO, USA) and are listed as follows.

### 4.4. Western Blot

The cell extracts were obtained after lysis with RIPA buffer (0.5% sodium deoxycholate, 50 mM Tris-HCl; pH 8, 150 mM NaCl, 1% NP40, 0.1% SDS) supplemented with protease and a phosphatase inhibitor cocktail (Roche, Basel, Switzerland), and equal amounts of protein were loaded onto SDS-PAGE gels. Following the transfer onto a nitrocellulose membrane, the blots were incubated overnight at 4 °C with the appropriate antibody, followed by incubation with a horseradish peroxidase-conjugated secondary antibody (1/2000, Cell Signaling, Danvers, MA, USA). The bands were visualized using the Clarity™ Western ECL substrate on Chemidoc systems (Bio-Rad, Hercules, CA, USA). The immunoblot analyses were performed using antibodies directed against P65 (8242T), P-P65 (3033S), IKKα (11930S), IκBα (4814T), and actin-HRP (5125S) (Cell Signaling). The PTX3 antibody (SC-373951) was purchased from Santa Cruz Biotechnology, Heidelberg, Germany. The protein expression was quantified by a densitometry analysis of the immunoblots using Image Lab software (1.53q), developed by Bio-Rad.

### 4.5. ELISA

The cells lines were plated in P60 at a density of 380,000 cells/well, in standard growth conditions for 72 h and treated with olaparib (20 µM), veliparib (10 nM), and talazoparib (20 µM). The media were then aspirated and replaced with serum-free medium for16 h incubation. The supernatants were collected, the particulates were removed by centrifugation and the protein concentration was measured using a BCA assay (Thermo Scientific, Waltham, MA, USA). The secreted VEGF and PTX3 in the cell culture supernatants were measured using the Human VEGF and Human Pentraxin 3/TGS-14 Quantikine ELISA kit (R&D Systems, Minneapolis, MN, USA).

### 4.6. Luciferase Assay

For the luciferase reporter gene assays, the cells were seeded in 12-well plates and transfected with NF-κB-luciferase and β-Galactosidase reporter plasmids, using the Fugene HD reagent (Promega, Madisson, WI, USA). Twenty-four hours later, the cells were lysed and the luciferase and β-galactosidase activities were measured using a luciferase assay (Promega, Madisson, WI, USA) and the Galacto Star system (ThermoFisher Scientific, Waltham, MA, USA), as described before [33]. The relative NF-κB promoter activity was calculated following the normalization against the β-galactosidase activity.

### 4.7. Human Angiogenesis Proteome Array

The cells lines were plated in P100 at a density of 1.5 × 10^6^ cells/well, in standard growth conditions and treated for 72 h with olaparib (20 µM), veliparib (10 nM), and talazoparib (20 µM). The media were then replaced with serum-free media for 16 h. The serum-free conditioned media were applied after the quantification with the BCA assay (Thermo Scientific) to the angiogenesis array membrane (Proteome Profiler Human Angiogenesis Array Kit; R&D System, Minneapolis, MN, USA), in accordance with the manufacturer’s instructions, to detect the relative levels of 55 angiogenic molecules. The quantification of the secretome-containing proteins was performed via the standard densitometry, using Chemidoc systems (Bio-Rad, Hercules, CA, USA), and the analysis was made using the Image Lab software developed by Bio-Rad.

### 4.8. Flow Cytometry and Cell Viability

The toxicity of olaparib (20 µM), veliparib (10 nM), and talazoparib (20 µM) was analyzed after 24 h, 48 h, and 72 h of treatments. Etoposide was used as a positive control (200 µM for 24 h). Annexin-V-PE (0.1 µg/mL; BD Biosciences) was used for the assessment of the phosphatidylserine exposure and therefore to detect the apoptotic cells 7AAD for the cell viability analysis. The assessments were performed in a Cytoflex (Beckman, Brea, CA, USA), and the data were analyzed using the Kaluza software 2.1.

### 4.9. Statistical Analysis

Each experiment was repeated at least three times, independently. All statistical analyses were performed with the Prism software v. 6.0 (GraphPad Software, San Diego, CA, USA). The averaged data were reported as means ± SEM. For comparisons between the two groups, an ANOVA test was used, and a multiple t-test was used to compare the columns. The statistical significance was accepted for *p* < 0.05. Symbols: * *p* < 0.05; ** *p* < 0.01; *** *p* < 0.001.

## 5. Conclusions

In summary, our study shows that PARP inhibitors inhibit the vasculogenic mimicry in breast cancer. We showed that PARPis inhibited the tube formation by modulating the NF-κB signaling pathway, resulting in a downregulation of PTX3. The NF-kB signaling and PTX3 provide biomarkers for a precision medicine approach to breast cancer and may represent potential druggable targets for the future, in combination with PARPis.

## Figures and Tables

**Figure 1 ijms-23-16171-f001:**
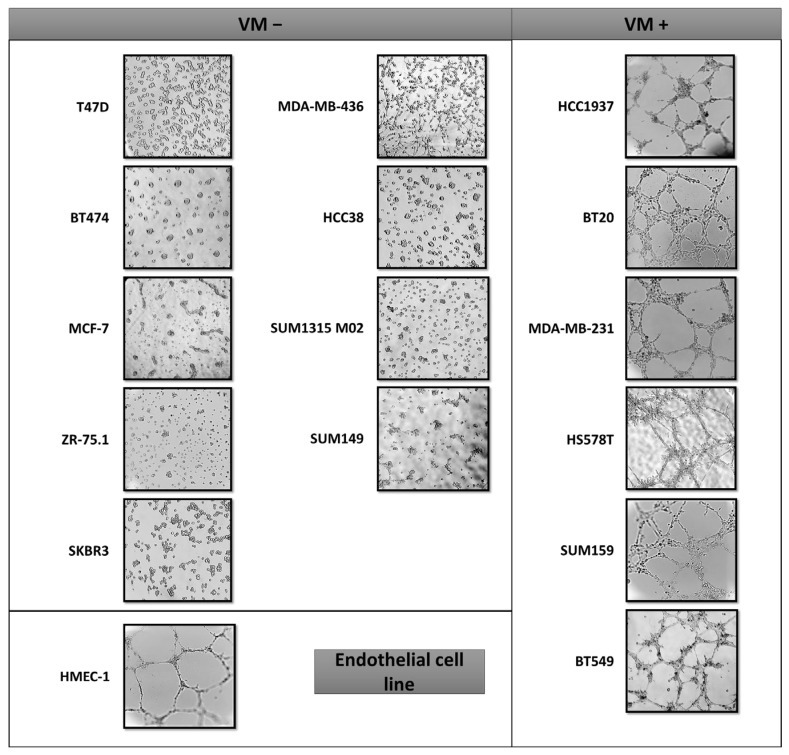
Vascular mimicry ability on the breast cancer cell lines subtypes. Phase contrast images show the vascular channel formation. Images were captured 24 h after plating on Matrigel. Experiments were realized at least 3 times.

**Figure 2 ijms-23-16171-f002:**
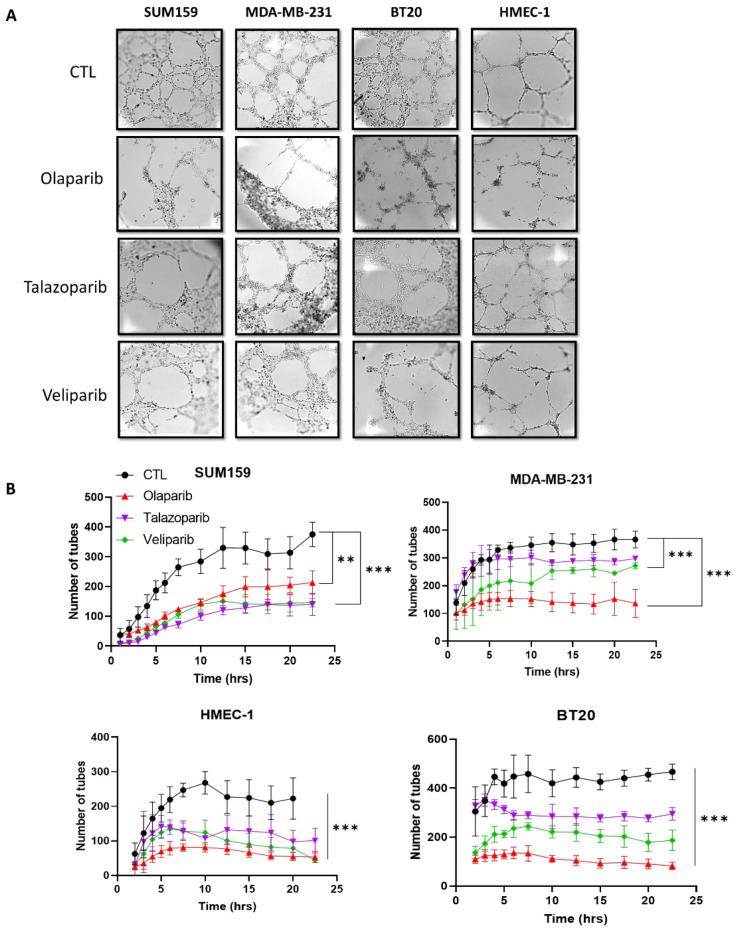
Effect of PARPis on the vascular mimicry. Time-lapse video microscopy experiments were realized on the HMEC-1 endothelial cells, and the breast cancer MDA-MB-231, SUM159, and BT20 cells treated for 72 h with olaparib (20 µM), talazoparib (10 nM), and veliparib (20 µM) before being seeded on Matrigel. (**A**) Representative pictures taken at 24 h. (**B**) Number of tubes formed. Experiments were realized at least three times and the data are expressed as mean ± SEM., ** *p* < 0.01, *** *p* < 0.001.

**Figure 3 ijms-23-16171-f003:**
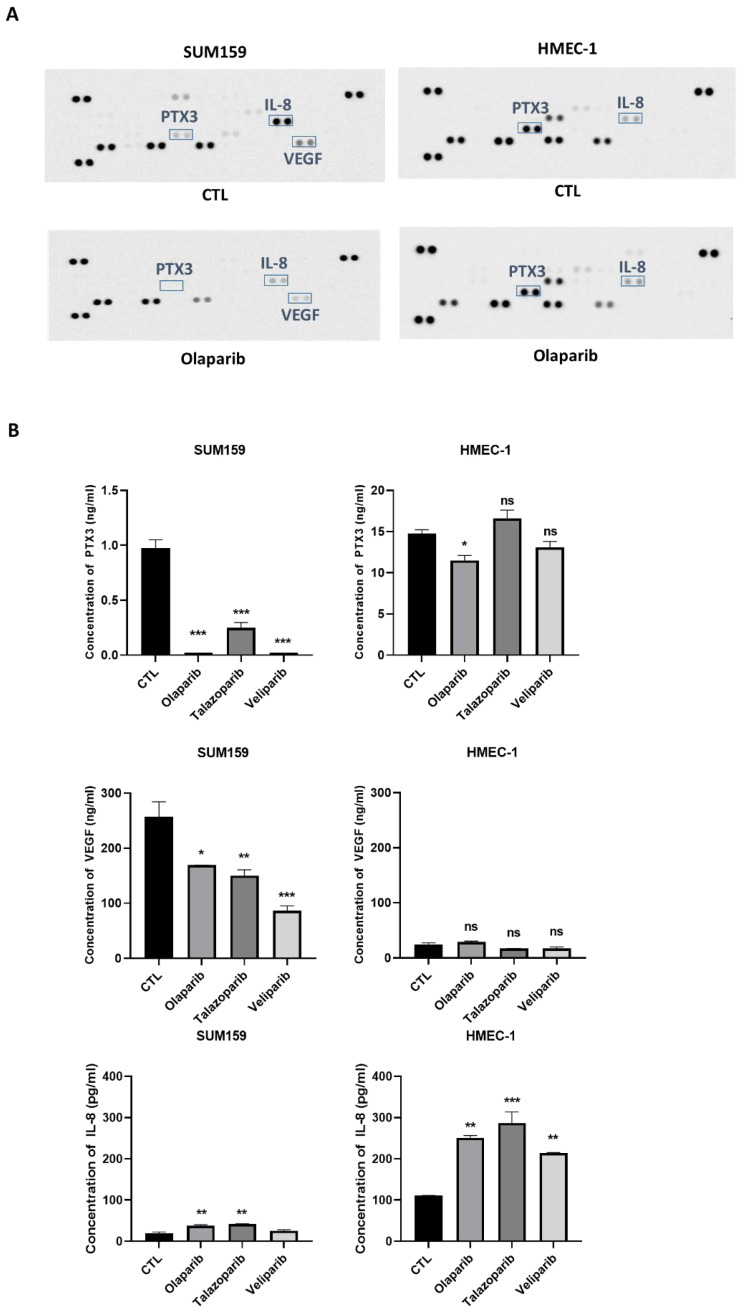
PARPi modifies the secretion of the angiogenesis-related proteins PTX3 and VEGF. (**A**) Cells were treated for 72 h with olaparib at 20 µM and, after the overnight incubation without SVF, a serum-free conditioned medium was applied onto the human angiogenic array membranes. The boxes indicate the major angiogenic regulated-molecules. (**B**) PTX3, VEGF, and IL-8 concentrations in the supernatant of SUM159 and HMEC-1, were measured by ELISA after 72 h of treatment by olaparib (20 µM), talazoparib (10 nM), and veliparib (20 µM). * *p* < 0.05; ** *p* < 0.01; *** *p* < 0.001; ns: no specific.

**Figure 4 ijms-23-16171-f004:**
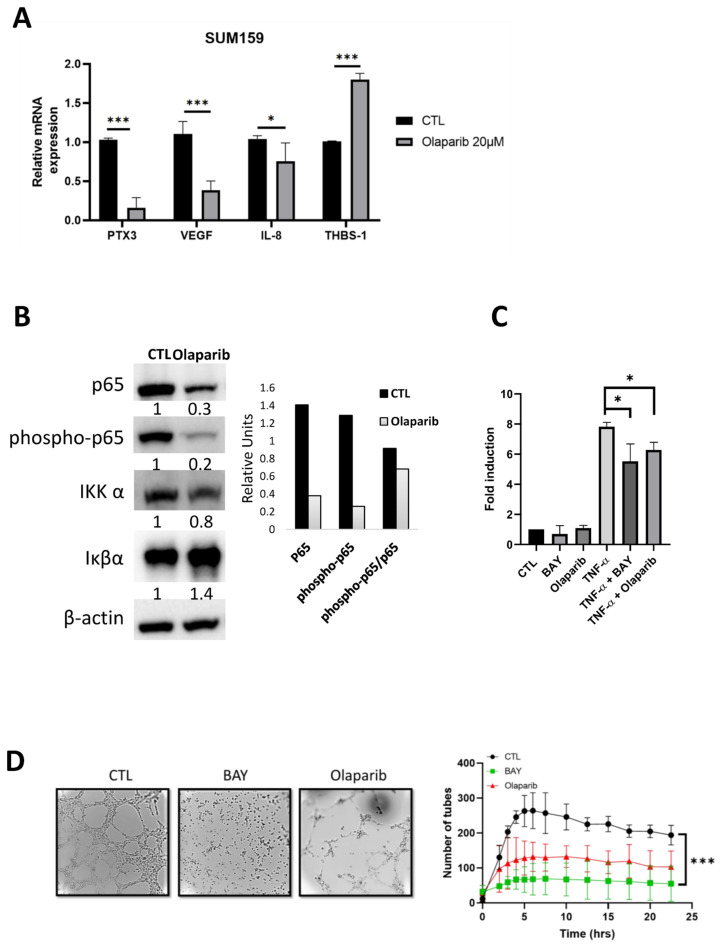
VM is regulated by PARPis through the NF-κB signaling. (**A**) RNA was isolated from the SUM159 cells and the gene expression was analyzed by the qRT-PCR. The results represent the relative transcript levels and are expressed as mean ± SEM from three independent experiments. (**B**) Western blot analysis of NF-κB p65 and the phospho-p65 protein, IKKα, and IκBα in the SUM159 cellular extracts, treated or not, with 20 µM olaparib for 90 min. Ratio of phospho-p65/p65 are shown as histograms after the quantification and normalization with β-actin. (**C**) SUM159 cells were transfected with plasmids encoding a NF-κB luciferase reporter gene and a plasmid encoding β-galactosidase. Cells were pre-treated for 5 h with the NF-κB inhibitor BAY-11-7082 (2.5 µM) or with olaparib (20 µM) and were then treated overnight with TNF-α (10 ng/mL). Data are average of the three biological replicates, each performed in triplicate. Fold induction, compared to the control condition. (* *p*  <  0.05, *** *p*  <  0.001). (**D**) Effect of the NF-κB inhibitor BAY-11-7082 on the tube formation assay after 24 h of treatment with BAY at 2.5 µM or 20 µM olaparib. Representative pictures were taken at 24 h and the tube formations were quantified. Experiments were realized at least three times and the data are expressed as mean ± SEM. * *p* < 0.05, *** *p* < 0.001.

**Figure 5 ijms-23-16171-f005:**
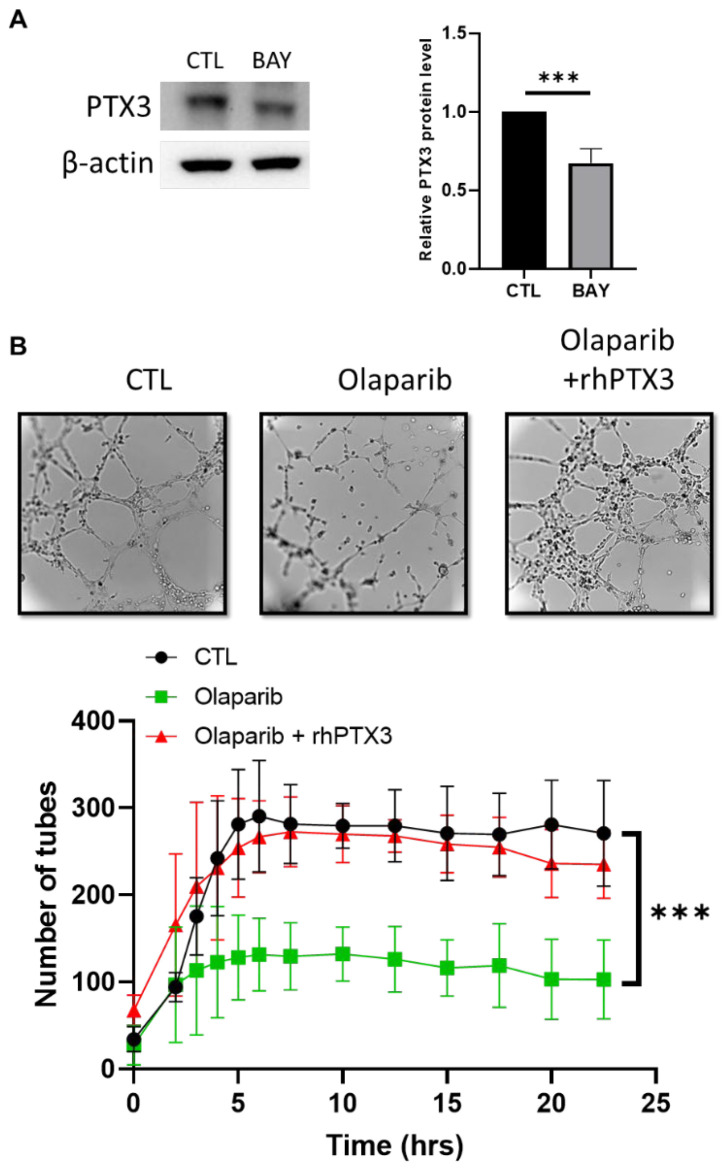
PTX3 reverses the PARPi effect on the VM formation. (**A**) Western blot analysis of PTX3 in the SUM159 cells after 24 h of treatment with BAY-11-7082 (2.5 µM). (**B**) Cells were treated for 72 h with olaparib (20 µM) and a human recombinant PTX3 was added for 24 h (1 µg/mL). Representative pictures were taken at 24 h and the tube formations were quantified. Experiments were realized at least three times and the data were expressed as mean ± SEM. *** *p* < 0.001.

**Table 1 ijms-23-16171-t001:** Characteristics of the breast cancer cell lines used in this study. Lum A: luminal A; Lum B: luminal B; IDC: invasive ductal carcinoma; AC: adenocarcinoma; DC: ductal carcinoma; AnC: anaplastic carcinoma.

Cell Line	Classification	ER	PR	HER2	Tumor	BRCA Status
T47D	Lum A	+	+	−	IDC	WT
BT474	Lum B	+	+	+	IDC	BRCA2
MCF-7	Lum A	+	+	−	IDC	WT
ZR-75.1	Lum A	+	+ −	−	IDC	WT
SKBR3	HER2	−	−	+	AC	WT
MDA-MB-436	Basal A	−	−	−	AC	BRCA1
HCC38	Basal B (Claudin Low)	−	−	−	DC	WT
SUM1315 M02	Basal B	−	−	−	IDC	BRCA1
BT549	Basal B (Claudin Low)	−	−	−	DC	BRCA1
SUM149	Basal B (Claudin Low)	−	−	−	DC	BRCA1
HCC1937	Basal A	−	−	−	DC	BRCA1
BT20	Basal A	−	−	−	IDC	WT
MDA-MB-231	Basal B	−	−	−	IDC	WT
HS578T	Basal B (Claudin Low)	−	−	−	IDC	WT
SUM159	Basal B (Claudin Low)	−	−	−	AnC	WT
HMEC-1	Endothéliale					WT

**Table 2 ijms-23-16171-t002:** List of primers used in the present study.

Target	Forward (5′ → 3′)	Reverse (5′ → 3′)
β-Actin	GGACTTCG AGCAAGAGATGG	AGCACTGTGTTGGCGTACAG
36B4	TCGACAATGGCAGCATCTAC	GCCTTGACCTTTTCAGCAAG
PTX3	CGTCTCTCCAGCAATGCAT	AAGAGCTTGTCCCATTCCGA
VEGF	ATGACCCAGTTTGGGAACA	TCCTGAATCTTCCAGGCAGT
uPA	TGCCCTGAAGTCGTTAGTGT	ATCTCCTGTGCQTGGGTGQQ
PAI1	CTCTCTCTGCCCTCACCAAC	GTGGAGAGGCTCTTGGTCTG
IL-8	TAGCAAAATTGAGGCCAAGG	AAACAAGGCACAGTGGAAC
TIMP-1	CTTCTGCAATTCCGACCTCGT	ACGCTGGTATAAGGTGGTCTG
THSB1	TTGTCTTTGGAACCACACCA	CTGGACAGCTCATCACAGGA

## Data Availability

Not applicable.

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
