# Peer review of "PARP Inhibitor Inhibits the Vasculogenic Mimicry through a NF-κB-PTX3 Axis Signaling in Breast Cancer Cells"

_ijms, 2022, doi:10.3390/ijms232416171_

Round 1
Reviewer 1 Report
Chivot et al., present an interesting study on the role of PARP signalling in basal triple negative breast cancer (TNBC) cells to regulate vasculogenic mimicry (VM). Using tube formation assays, they show convincing evidence that PARP inhibition impairs VM tube formation. This correlates with decreased expression of angiogenic factors which are known to lie downstream from NFKB signalling and a corresponding decrease in NFKB-regulated phosphorylation events. The authors propose a mechanism whereby PARPi inhibits signalling through the NFKB pathway to decrease expression of an angiogenic factor PTX3. Overexpression of recombinant PTX3 rescues the effect of PARPi on VM in SUM159 TNBC cells.
The authors demonstrate a clear correlation between NFKB signalling, PTX3 expression and PARP inhibition but it is less clear whether this is a direct or indirect effect. Inhibition of PARP with Olaparib appears to have a much greater effect on PTX3 expression (Fig 3 and 4a) compared to NFKB inhibition (Fig 5a). What happens when you inhibit both PARP and NFKB signalling? Is the effect additive or epistatic? Is the effect of PARPi on PTX3 expression still seen when PARP is inhibited with Talazoparib or Veliparib given that both these inhibitors had comparable effects on VM in SUM159 cells? And given the variability and debate about the link between effects of PARPi on NFKB signalling (raised in the discussion) do you see similar changes in NFKB signalling or PTX3 expression changes in any of your other Olaparib-sensitive cell lines (eg MDA-MB-231, BT20)?
It would be helpful to include a brief description of NFKB signalling and PTX3 in the introduction. As a reader, it was a long wait to find out what PTX3 was and why it was interesting… information which currently only comes at the end in the discussion section.
Minor points:
Figure 2 – Please clarify the quantification of VM. The graphs are labelled as “Number of tubes” but the figure legend refers to them as “quantification of length of tube formed”. I’m not familiar with these measurements – what exactly is being measured here?
Page 4, line 110 – The statement that “Olaparib seems to have greater potency than Talazoparib and Veliparib” is a bit ambiguous. At face value, I understand what the authors are saying but without evidence of the IC50 values for each inhibitor in the cell lines used, it’s hard to call the relative efficacy of PARP inhibition under these experimental conditions. Olaparib and Veliparib were used at (20uM) and Talazoparib at (10nM). It is known that PARP inhibition by Talazoparib > Olaparib > Veliparib using in vitro IC50 values (https://doi.org/10.1073/pnas.2121979119) but whether the concentrations used in these assays and cell lines are directly comparable to one another in terms of PARP inhibition is not clear – so the authors should not attempt to make statements about relative potency.
Page 1, line 30 – typo “cancer along women” should be “cancer among women”
Page 2, line 84 – “all basal cell lines do not form VM” should read “all luminal cell lines do not form VM”
Page 3, Table 1 – typo “BRCA statut” heading should read “BRCA status”
Author Response
We thank the reviewer for these positive comments and pointing out some mistakes.
The authors demonstrate a clear correlation between NFKB signalling, PTX3 expression and PARP inhibition but it is less clear whether this is a direct or indirect effect. Inhibition of PARP with Olaparib appears to have a much greater effect on PTX3 expression (Fig 3 and 4a) compared to NFKB inhibition (Fig 5a). What happens when you inhibit both PARP and NFKB signalling? Is the effect additive or epistatic? Is the effect of PARPi on PTX3 expression still seen when PARP is inhibited with Talazoparib or Veliparib given that both these inhibitors had comparable effects on VM in SUM159 cells? And given the variability and debate about the link between effects of PARPi on NFKB signalling (raised in the discussion) do you see similar changes in NFKB signalling or PTX3 expression changes in any of your other Olaparib-sensitive cell lines (eg MDA-MB-231, BT20)?
The relationship between the effects of PARP and NF-kB signaling is pursued, therefore the inhibition of these 2 signaling is in progress and is the subject of a new work.
As shown in figure 3B, the concentration of PTX3 is also inhibited when tumor cells are treated with talazoparib or veliparib.
As you can show in the figure below, olaparib induces changes in NFKB signaling and PTX3 expression in MDA-MB-231 cell line
It would be helpful to include a brief description of NFKB signalling and PTX3 in the introduction. As a reader, it was a long wait to find out what PTX3 was and why it was interesting… information which currently only comes at the end in the discussion section.
As suggested by the reviewer, we have added a brief description of NFKB signaling and PTX3 in the introduction.
Minor points:
Figure 2 – Please clarify the quantification of VM. The graphs are labelled as “Number of tubes” but the figure legend refers to them as “quantification of length of tube formed”. I’m not familiar with these measurements – what exactly is being measured here?
The figure's legend has been corrected as “Number of tubes”. A tube was defined as a linear sequence of cells linking two nodes and tube formation was quantified using ImageJ software and the plugin Angiogenesis Analyzer
Page 4, line 110 – The statement that “Olaparib seems to have greater potency than Talazoparib and Veliparib” is a bit ambiguous. At face value, I understand what the authors are saying but without evidence of the IC50 values for each inhibitor in the cell lines used, it’s hard to call the relative efficacy of PARP inhibition under these experimental conditions. Olaparib and Veliparib were used at (20uM) and Talazoparib at (10nM). It is known that PARP inhibition by Talazoparib > Olaparib > Veliparib using in vitro IC50 values (https://doi.org/10.1073/pnas.2121979119) but whether the concentrations used in these assays and cell lines are directly comparable to one another in terms of PARP inhibition is not clear – so the authors should not attempt to make statements about relative potency.
We agree with this comment and we have deleted the statement.
Page 1, line 30 – typo “cancer along women” should be “cancer among women”
Along is corrected to among
Page 2, line 84 – “all basal cell lines do not form VM” should read “all luminal cell lines do not form VM”
We have corrected this mistake
Page 3, Table 1 – typo “BRCA statut” heading should read “BRCA status”
The table is corrected
Reviewer 2 Report
In the paper titled: "PARP Inhibitor inhibits vasculogenic mimicry through a NF-κB-PTX3 axis signaling in breast cancer cells" the authors assess if treatment with Olaparib, Talazoparib, and Veliparib inhibits vascular channel formation of breast cancer cell lines. They found that PARP inhibitors act as potent inhibitors of VM formation in triple negative breast cancer cells independently of BRCA status. The paper is important in that it shows PARP inhibitors mechanism of action. The paper is well written, and the data is convincing. Also, the movies are beautiful. Below are a few comments:
-Do not use PARPi in plural (PARPis), as it is already plural. Consider using the standard nomenclature PARP inhibitor.
-In table 1, please correct statut-> status
Author Response
We thank the reviewer for these positive comments and pointing out some mistakes.
-Do not use PARPi in plural (PARPis), as it is already plural. Consider using the standard nomenclature PARP inhibitor.
We have corrected all PARPis par PARPi
-In table 1, please correct statut-> status
The table is corrected
Round 2
Reviewer 1 Report
The authors have made satisfactory changes to the introduction and minor corrections. However, the vast majority of my major concerns have not been addressed or attempted so my opinion remains unchanged. All of the previous questions raised could be answered using existing lab techniques and in my opinion form part of this study and not the next. I would like to see some attempt to address the experimental questions raised in my first review.
Round 3
Reviewer 1 Report
Thank you for the updated response and clarity. It is good to see that the effect of Olaparib treatment to decrease NFKB phospho-p65 and PTX3 levels in the MDA-MB-231 cell line is consistent with previous data (albeit to a more modest extent).
It’s also good to see that the authors agree that the relationship between PARPi and NFKB signalling in PTX3 regulation is important. The reason I asked for the co-drug treatment to elucidate whether PARPi and NFKBi was epistatic or additive - is that without this data the manuscript draws most of the conclusions about the role of PARPi and NFKB signalling from correlative data. PARPi correlates with decreased NFKB signalling which correlates with decreased PTX3 expression. While overexpression of PTX3 can rescue the effects of PARPi – the relationship between PARPi treatment and NFKB signalling is more complex and hasn’t been fully explored. Since correlation doesn’t necessarily imply causation, it makes it harder to justify a simplistic conclusion about signalling pathways.
If there is strong reluctance to include the drug combination data that might allow a stronger conclusion to be drawn, perhaps due to time constraints,– perhaps the solution is to include a version of the comments made in review (copied in below) in the discussion for transparency.
The relationship between the effects of PARP and NF-kB signaling in the PTX3 regulation is pursued. As PARP1 and p65NF-kb have been shown to interact and modulate gene expression (Stanisavljevic et al, J Cell Sci. 2011) therefore investigating this interaction in PTX3 gene expression and furthermore the inhibition of these 2 signaling as suggested by the reviewer is of great importance to elucidate mechanisms regulating PTX3 functions. Moreover, some data combining PARPi and NF-kb showed an increase efficacy of the combination of two drugs (Li et al, Mol.Cancer.Res., 2019) and also could be observed in the VM process.
Author Response
We thank the reviewer for all his comments and as suggested we have added a comment and 2 references concerning the drug combination in the conclusion to support this comment.